# An Epigenetic Perspective on Intra-Tumour Heterogeneity: Novel Insights and New Challenges from Multiple Fields

**DOI:** 10.3390/cancers13194969

**Published:** 2021-10-03

**Authors:** Sven Beyes, Naiara Garcia Bediaga, Alessio Zippo

**Affiliations:** Laboratory of Chromatin Biology & Epigenetics, Department of Cellular, Computational and Integrative Center for Integrative Biology (CIBIO), University of Trento, 38123 Trento, Italy; sven.beyes@unitn.it (S.B.); naiara.garciabediaga@unitn.it (N.G.B.)

**Keywords:** cancer, epigenetics, intra-tumour heterogeneity, single-cell OMICs, epigenetic heterogeneity, metastasis

## Abstract

**Simple Summary:**

Although research on cancer biology in recent decades has unveiled the main genetic perturbations driving the onset of tumorigenesis, we are still far from properly treating this disease without the occurrence of drug resistance and metastatic burden. This achievement is hampered by the onset of intra-tumour heterogeneity (ITH), which increases cancer cell fitness and plasticity, thereby fostering cell adaptation to foreign environments and stimuli. In this review, we discuss the contribution of the epigenetic factors in sustaining ITH and their interplay with the tumour microenvironment. We also highlight the recent technological advancements that are contributing to defining the epigenetic mechanisms governing tumour heterogeneity at the single-cell level.

**Abstract:**

Cancer is a group of heterogeneous diseases that results from the occurrence of genetic alterations combined with epigenetic changes and environmental stimuli that increase cancer cell plasticity. Indeed, multiple cancer cell populations coexist within the same tumour, favouring cancer progression and metastatic dissemination as well as drug resistance, thereby representing a major obstacle for treatment. Epigenetic changes contribute to the onset of intra-tumour heterogeneity (ITH) as they facilitate cell adaptation to perturbation of the tumour microenvironment. Despite being its central role, the intrinsic multi-layered and reversible epigenetic pattern limits the possibility to uniquely determine its contribution to ITH. In this review, we first describe the major epigenetic mechanisms involved in tumourigenesis and then discuss how single-cell-based approaches contribute to dissecting the key role of epigenetic changes in tumour heterogeneity. Furthermore, we highlight the importance of dissecting the interplay between genetics, epigenetics, and tumour microenvironments to decipher the molecular mechanisms governing tumour progression and drug resistance.

## 1. Introduction

Cancer is characterized by a high degree of heterogeneity that occurs at both the inter- and intra-tumour levels [1]. Inter-tumour heterogeneity defines differences between tumours of the same origin across different patients [2], and it lays the foundation of the classification of tumours in subtypes that are often used to guide treatment decisions. Tumour subtypes show specific genetic alterations, molecular signatures, tumourigenic pathways and biological behaviours, resulting in different clinical outcomes [3]. Intra-tumour heterogeneity (ITH), on the other hand, refers to the existence of distinct subpopulations of cancer cells within individual tumours. Since intra-tumour heterogeneity maximizes the fitness of the cancer cell population in dynamic tumour environments, it is likely to govern key aspects of tumour biology including tumour growth, metastasis, and drug resistance and thus constitutes a major obstacle to effective and individualized cancer treatment. In addition, ITH can lead to molecular misclassification [4,5,6] due to the presence of multiple cell populations belonging to different molecular groups according to standard classifications within a single biopsy.

ITH arises from at least three mechanisms, which are not mutually exclusive and act together to create a complex system with multiple layers of heterogeneity: (1) genetic heterogeneity, due to stochastic accumulation of mutations that results in tumours acquiring subclones with distinct genotypes during tumour evolution; (2) non-genetic heterogeneity from variations in regulatory mechanisms, including epigenetic, posttranscriptional, and posttranslational modifications, among others; (3) tumour microenvironmental (TME) heterogeneity, due to selective pressures in distinct regions of the tumour [7]. Varying TME factors include hypoxia, tissue stiffness, nutrient availability, and cell–cell interactions with other malignant and non-malignant cells such as infiltrating immune cells, stromal cells, and endothelial cells. 

Heterogeneous tumour ecosystems have traditionally been profiled using bulk experimentation studies that only revealed an averaged cellular behaviour and thus have heavily underestimated the complex intra-tumour heterogeneity that characterizes cancer samples. With the advent of single-cell multi-omics technologies, cancer research has shifted to a new paradigm that has facilitated a comprehensive analysis of the various cellular programs within human tumours at single-cell resolution and thus improved our understanding of tumour heterogeneity and cancer evolution.

In this review, we discuss the contribution of epigenetic alterations to tumour heterogeneity and their interplay with other sources of ITH. In addition, we provide an overview of the novel approaches that can be applied to dissect the complexity of the molecular mechanisms governing cancer-specific epigenetic changes at a single-cell level.

## 2. Routes toward Epigenetic Heterogeneity

In multicellular organisms, different cell types arise from the same cell of origin through a hierarchical differentiation process that specifies cell functions and tissue localisations. This specified, yet diversified tissue organization is disrupted in tumour cells, resulting in a heterogeneous phenotypic pattern. In 1976, Nowell proposed that cancer development is driven by clonal evolution, which results in adapted clones driving heterogeneity [8]. One year later, Fidler and Kripke showed that clones derived from a murine melanoma model differed in their appearance and the ability to form metastases [9], indicating that functional heterogeneity is “inherited” in these cancer cells. In the following years, more evidence was gathered to describe this heterogeneity and its origin, leading to two different hypotheses: (1) the cancer stem cell (CSC)-centred view, in which a tumour arises from a single cell of origin, which then gives rise to different clones, and (2) the clonal-selection-derived model, driven by a stepwise acquisition of mutations due to which different tumour cell subclones can arise in parallel (branched evolution) [10]. To prove the branched evolution theory, Gerlinger and colleagues analyzed multiple spatially separated renal carcinoma samples from the same patient and showed that only a subset of the mutations discovered were present in all the analyzed samples [11]. The same was shown for a cohort of breast cancers, in which different tumour subpopulations showed a diversity of mutational spectra, linked to metastasis and tumour recurrence [12]. 

These findings indicate that the information retrieved from a single patient biopsy might not be sufficient to determine the optimal first-line treatment. This can result in incomplete tumour eradication and increased selective pressure for other cancer cell subpopulations, leading to increased aggressiveness and resistance of the remaining clones and subsequent failure of the second-line treatment.

Over the last 30 years, an increasing amount of evidence has been gathered showing that cancer development was driven by not only genetic but also by non-genetic alterations linked to epigenetic dysregulation, including changes in the 3D genome organization, chromatin state, and DNA methylation. In comparison to the mutational load, which is considered a vertical transfer and involves cell division and the subsequent survival of the daughter cells, epigenetic changes are dynamic and reversible, allowing cancer cells to adapt to novel environmental stimuli, such as nutrient deprivation, oxidative stress, and microenvironment-related signals [13,14]. Since 2006, it has been proposed that epigenetic changes can contribute to the onset of tumorigenesis by altering the expression pattern of ”tumour progenitor genes”, which are interconnected with epigenetic modifiers and modulators, whose function is perturbed by the TME and genetic alteration [15]. Multi-omics approaches are nowadays revealing that the epigenetic patterns of tumour samples result from a high degree of epigenetic ITH (eITH) [16,17,18]. To decipher the influence of the environmental factors on the eITH of tumours, it is fundamental to perform single-cell analyses. This approach applied to cell populations isolated from breast cancer patients derived pre- and post-neoadjuvant chemotherapy showed different phenotypes, although the genetic pattern and mutational load detected did not change between the two time points [19]. In addition, the authors showed that the treatment decreases certain subpopulations within a tumour type, while an increase was observed for other subpopulations. Despite their relevance, these studies are still limited in determining the functional link between the TME and the epigenetic changes occurring among cancer cells during tumour progression. Understanding and deciphering the eITH of cancer samples will help to further personalize the treatment regimens targetting all cells of a tumour and taking into account the possible resistance mechanisms already present within the cancer cell population.

### 2.1. Impact of Epigenetic Heterogeneity on Tumour Progression and Metastasis

Multiple layers of epigenetic control can be corrupted during cancer progression and metastasis. By analyzing the different patterns of epigenetic alterations in multiple cancers, it has been shown that epigenetic heterogeneity represents a hallmark of cancer cells. Diverse routes drive towards epigenetic aberration during tumour progression and metastasis formation, giving rise to eITH, which increases cancer cell plasticity (Figure 1). However, it remains difficult to dissect the contribution of individual alterations and their interplay in determining the overall epigenetic heterogeneity that emerges by analyzing the epigenome of cancer cells as a bulk.

#### 2.1.1. DNA Methylation

The first epigenetic marker extensively studied in the context of cancer was DNA methylation. DNA methylation occurs at cytosine-guanine sites, and in healthy tissues a large portion of the genome (60–80%) is methylated, ensuring genome stability and silencing of repetitive elements within the heterochromatic regions. However, in cancer cells, a large fraction of the genome actually shows a hypomethylated state compared to the healthy controls, with enrichment at repetitive elements (RE) [20,21]. Hypomethlyation of RE can foster a higher rearrangement during cell division, driving tumour progression [22]. On the contrary, induced hyopmethylation by treatment with a DNA demethyltransferase inhibitor (DNMTi) can also be exploited for cancer therapy, as was shown for epithelial cancers [23,24], by demethylation and expression of endogenous retrovirus genes, triggering a type I interferon response, allowing for immune checkpoint therapy. In addition, gene promoters are characterized by CpG rich regions (termed CpG islands) that are characterized by a low DNA methylation level, favouring chromatin accessibility to transcription factors (TF) that regulate gene expression. In cancer, many promoters of potent tumour suppressor genes, such as CDKN2A, whose gene product p16INK4a is important for cell cycle control, and the DNA repair gene MGMT, are methylated and the genes are downregulated [25,26]. In an elegant study, Brock et al. showed that DNA methylation in different prostate cancer and adjacent tissue samples reflects the genetic origin of the tumours, highlighting the power of using epigenetic markers to stratify tumour heterogeneity [27]. When analyzing the distribution of DNA methylation patterns in the tumour samples, they could identify a high level of intra-tumour heterogeneity in the vicinity of prostate-specific gene regulatory elements, which might be linked to intra-tumoural gene expression heterogeneity. A similar approach was employed by Liu and colleagues, who used a pan-cancer analysis to further determine the genome-wide pattern of DNA methylation and its relative enrichment at specific sites [28]. Indeed, they showed that cancer-specific methylation is overrepresented at TF-encoding genes, resulting in extensive changes in their expression, triggering transcriptional reprogramming of these cells. 

Besides the methylation of tumour suppressor genes, DNA methylation can also affect TF binding, as was shown for hypoxia-induced TF [29] and 3D genome organization [30]. For example, the TF CCCTC-binding factor (CTCF) binds to a CpG-rich DNA binding motif at regulatory elements, where it acts as an insulator by favouring chromatin looping. Methylation of its DNA binding motif strongly reduces CTCF binding affinity, thereby affecting the function of the regulatory element harbouring the methylated CTCF binding site [31]. Flavahan and colleagues demonstrated that human IDH mutant gliomas exhibit hypermethylation at CTCF-binding sites, resulting in alteration 3D structure genome-wide and activation of the oncogene PDGFRA by allowing a constitutive enhancer to interact aberrantly with PDGFRA promoter [32]. Notably, treatment of IDH mutant glioma cells with a demethylation agent reduced methylation at CTCF binding sites and partially restored CTCF binding and insulator function, which in turn restored PDGFRA expression [32].

#### 2.1.2. Histone Modifications and Histone Variants

Besides DNA methylation, histone posttranslational modifications (PTM) play a central role in establishing different chromatin contexts that can modulate both genetic and non-genetic genome functions [33,34]. The combination of histone PTMs and chromatin players (including writers, erasers, readers of histone modifications, and chromatin remodelers) define the heritable epigenome of cells, yet ensuring flexibility in gene expression by dynamic and reversible changes that can occur at specific regulatory regions, in response to upcoming signals. In healthy cells, a balance between specific modifications and chromatin players is ensured to maintain the epigenetic state. In corrupted cells such as cancer, an imbalance between modifications and chromatin players can occur, resulting in gene expression changes, deregulated cell growth, and fast response to treatment. This imbalance can be due to mutations affecting genes codifying for the chromatin players, which are responsible for a high percentage of overall cancer-driver mutations [35], or due to the dysregulation of their expression pattern. Apart from dissecting the mutational spectrum of multiple chromatin players in different tumour types [36,37], we are still far from fully understanding their contributions to tumour progression or drug resistance. As an example, it has been shown that the sole profiling of histone modifications could be used for the prediction of prostate cancer recurrence, without further analysis of the subsequent regulatory changes [38,39]. Besides changes in the histone modifications, it has been recently described that the relative abundance of histone variants or linker histones play a central role in defining the epigenetic pattern of cancer cells. A hallmark study demonstrated that tumours show heterogeneous levels of the linker histone H1.0, whose chromatin-associated levels correlated with the tumour differentiation status, and an increase in H1.0 results in a decreased cell proliferation of single cells [40]. For ovarian cancer, a change in expression of different H1 variants has been observed during the progression from adenoma to adenocarcinoma [41], allowing tumours to be classified on the basis of the H1 variants’ expression levels. In prostate tissue, the staining of the somatic subtype H1.5 could be used to separate prostate cancer cells from adjacent benign prostatic epithelium [42]. In a recent study, Yusufova and colleagues showed the effect of histone variants on the 3D genome architecture, as a functional loss of Histone H1 results in global chromatin decompaction and the reshaping of chromatin compartmentalization [43].

On the other hand, mutations in histone genes can alter the epigenetic landscape by mimicking a certain histone PTM and make histones insensitive to other PTMs on the mutated locus. These mutations have been detected mainly in genes encoding histone H3 variants, such as a Lys-27-Met mutation in genes coding for histone 3.3 variants (H3K27M), identified in paediatric high-grade gliomas (HGG) [44]. This mutation results in a decrease in repressive H3K27me3 and a redistribution of H3K27me2 [45] by mimicking loss of PRC2 function, as was shown in Drosophila [46]. In addition, in HGG, it was shown that regions decorated with H3K27M showed an increase in H3K27ac, a mark for active enhancers and promoters [47,48]. This active chromatin state leads to the expression of previously silenced genes and might also alter the 3D genome conformation. The discovery of the H3.3G34 variant shed light on the impact of a histone modification in *cis*, as it affects the methylation of the neighbouring H3.3K36 residue [49]. The initial discovery of oncohistones and the increasing evidence of their implications not only in paediatric tumours are adding another layer of complexity to the epigenetic dissection of tumours [50]. 

#### 2.1.3. Chromatin Accessibility 

Nucleosome occupancy at both distal and proximal *cis*-regulatory elements hampers gene regulation by limiting the access of TFs and chromatin factors to the cognate DNA elements, thereby representing a major barrier for transcription to occur. Nucleosomal remodeling is mainly driven by multi-subunit complexes, termed chromatin remodeling complexes, composed of different protein families with the most prominent member in the context of cancer being the mSWI/SNF complex. These ATP-dependent remodeling factors disrupt the connection between nucleosomes and DNA by hydrolysis, allowing them to deposit, eject, or slide nucleosomes, thereby altering the chromatin structure and ultimately allowing for the establishment of an active chromatin state [51]. In >20% of cancers, mutations in genes coding for mSWI/SNF remodeling factors have been identified [52], resulting in the perturbation of the protein function and changing the global chromatin accessibility pattern. In a pioneering study, the biallelic inactivation of SMARCB1, the core BAF complex subunit, resulted in a global decrease in enhancer accessibility, which could be restored by the re-expression of wild-type SMARCB1 [53], linking the loss of a remodeling complex subunit to global changes in accessibility patterns. To remodel nucleosomes, the complexes have to be recruited to the specific loci. This recruitment is facilitated by sequence-specific pioneer TF, which can bind their cognate DNA sequences on closed chromatin and recruit the remodelers to increase accessibility. 

In a hallmark study, Alonso-Curbelo and colleagues showed that injury in pancreatic tissue harbouring a Kras mutation drives the establishment of an epigenetic state more resembling pancreatic ductal adenocarcinoma (PDAC) than healthy pancreatic tissue [54]. Interestingly, the changes in accessibility could be detected shortly after injury. The newly accessible regions showed enrichment for motifs of pioneer TFs and a pattern similar to PDAC samples, indicating a rapid reprogramming of the affected cells towards a cancerous phenotype of injured pancreatic tissue. However, to further verify the results described, the direct binding of the identified TFs to the target regions should be verified, and a supporting investigation of the remodeling factors involved and the histone PTM patterns allowing for de novo chromatin accessibility should be conducted. This will further contribute to the understanding of how this rapid change of chromatin accessibility occurs and which cell-type-specific drivers orchestrate it. 

### 2.2. Epigenetic Reprogramming in Cancer

Cancer cells can respond in different ways to environmental stimuli, which often only affect a fraction of the whole cancer bulk. In response to stimuli, master TFs are often deregulated, which in turn have a great impact on cancer cell phenotypes. It has been proposed that TFs play a central role in driving epigenetic reprogramming, by suppressing the expression of lineage-specific transcriptional programs or by de novo activation of *cis*-regulatory elements, which ultimately result in an altered epigenetic program. Although the mechanisms and players that drive this epigenetic reprogramming are not yet completely understood, some players have already been identified and linked to cancer heterogeneity and tumour progression. 

#### 2.2.1. Transcription Factor-Mediated Reprogramming

TFs play an important role in cancer progression, as they can alter the epigenetic landscape of single cells by modulating chromatin accessibility at *cis*-regulatory elements, favouring the establishment of an active (or repressive) chromatin state and ultimately changes in the gene expression pattern. In small-cell lung cancer (SCLC), copy number alteration-driven upregulation of the TF Nfib fosters the activation of evolutionary conserved *cis*-regulatory elements in primary tumour cells. These activated enhancers stimulate the expression of neuronal gene expression programs, which support metastasis burden [55]. Of importance, epigenome profiling of metastatic cells clearly distinguished them from the primary tumours, and these changes were related to the altered Nfib activity and expression levels. However, it has not been clarified whether other genetic or epigenetic alterations contributed to establishing the metastatic-specific epigenetic pattern. For PDAC, Roe and colleagues could show that that the expression of the pioneer factor FOXA1 in primary pancreatic tumour organoids is sufficient to orchestrate global enhancer reprogramming and shifting the tumours towards a more aggressive and disseminating phenotype [56] without being affected by additional driver mutations. Most of the H3K27ac GAIN regions (92.9%) were located in introns and intergenic loci, indicating that the major part of the reprogramming occurred at distal *cis*-regulatory elements.

In another study, the overexpression of the oncogene c-MYC in immortalized human mammary epithelial cells resulted in an epigenetic reprogramming and a repression of lineage-specifying TFs. At the same time, MYC overexpression caused the reactivation of developmental pathways and the switch of the cancer cells towards a more aggressive stem-cell-like state [57,58,59]. In the same tumour context, the role of a SOX10^high^ subpopulation within a tumour bulk was highlighted. In the mammary cell hierarchy, SOX10 plays an important role in the maintenance of multi-lineage potential during mammary gland development. During tumour progression, the SOX10^high^ subpopulation of the tumour showed a higher level of dedifferentiation and features of epithelial-to-mesenchymal transition (EMT), linking subpopulation-specific TF activity to the aggressiveness of cancer [60]. 

Besides the direct deregulation of TFs, multiple signals provided by the TME can support epigenetic reprogramming, fostering cancer cell plasticity and tumour progression. Among these, TGF-β signaling plays a central role in the induction of EMT, a reversible process important for cell migration [61]. In healthy tissue, paracrine TGF-β acts as cytostatic factor by repressing cell proliferation, while in certain tumourigenic contexts its activity can change dramatically. TGF-β can trigger migration and subsequent invasion of cells by an upregulation of EMT-inducing TFs such as SNAIL, SLUG, TWIST1/2, and ZEB1/2 [62,63,64]. These inducers can trigger global epigenetic changes by recruiting transcriptional repressors such as the histone demethylase LSD1 [63] and PRC2 components such as EZH2 [65] or by interfering with the binding of transcription cofactors, such as p300 [66]. The induced changes result in the loss of the epithelial phenotype and a shift towards a more mesenchymal state. Given the gradient in TGF-β availability around the tumour mass, only a subpopulation of cells will experience this reprogramming towards a more migratory phenotype, leading to the dissemination of cells and eventually metastasis formation. For squamous cell carcinoma (SCC), it was shown that the epigenetic background and the transcriptional state of individual tumour-initiating cells can affect the priming for EMT induction, clearly indicating that primed regions in a subpopulation at an earlier stage might have a strong influence on the outcome of extracellular factors at a later stage [67]. The authors showed that the epigenetic priming of the tumour cells of origin has a strong impact on the development of different tumour cell types independently of the driver mutations.

Besides the overexpression of oncogenes or the activation of TFs in response to deregulation of external stimuli, cancer treatments can also have a striking effect on the epigenetic landscape. In recent years, it became evident that drug-mediated epigenetic reprogramming imposes a major challenge for drug administration and cancer treatment, as long-term drug exposure can result in unwanted effects. Bi and colleagues showed that in luminal breast cancer the acquisition of endocrine resistance is linked to the reshaping of the enhancer activity, altering the transcriptional program [68]. Mechanistically, it has been proposed that the assembly of oncogenic TFs reorganized the enhancer landscape, promoting the activation of an endocrine resistance-associated transcriptional program. 

Comparable results were obtained by using vemurafenib, which triggered a stepwise reprogramming of a subpopulation of melanoma cells towards a resistant phenotype with an altered chromatin accessibility landscape identified by ATAC-seq, which led to dedifferentiation and subsequent activation of novel, until then silenced, pathways [69]. Sharma and colleagues used the non-small-cell lung carcinoma (NSCLC) cell line PC9 to show that treatment with receptor tyrosine kinase-inhibitors (RTKi) results in incomplete depletion of tumour cells, as a small subpopulation was not affected by the treatment [70]. Further analysis of this subpopulation revealed a decrease in H3K4me3 and H3K4me2, which was linked to an upregulation of the histone demethylase KDM5A. Comparable results were obtained in a study conducted in Glioblastoma stem cells treated with the RTKi dasatinib [71]. In this context, the resistant slow-dividing cells were characterized by an upregulation of the Notch signaling pathway, which resulted from the derepression of *cis*-regulatory elements controlling the expression of NOTCH target genes. Specifically, it has been shown that the KDM6 family H3K27 demethylases play a major role as drivers of enhancer reactivation [71]. Similar results were obtained in a model of ovarian cancer, in which cells resistant to cisplatin showed the redistribution of super-enhancers (SE), which enables them to survive the treatment regimen and increase malignancy over time [72]. Overall, these results clearly demonstrate that cancer cells can respond and adapt to multiple external cues by reprogramming their epigenetic landscape either through the direct activity of deregulated TFs or by the recruitment of chromatin players. Although the interplay between genetic alterations, environmental signals and the epigenetic state represents a major driver of cancer cell plasticity and tumour heterogeneity (Figure 1b), our understanding of the mechanisms guiding these orchestrated actions is still limited. The implementation of technological platforms to measure in the same cells both the genetic and epigenetic changes, coupled with the analysis of the transcriptional program, would allow us to determine the contribution of epigenetic reprogramming to tumour progression and drug resistance. 

#### 2.2.2. Changes in 3D Genome Organization

The eukaryotic genome is packaged into a highly organized hierarchical structure that is directly associated with the establishment and maintenance of regulatory networks and transcriptional regulation. Nucleosomes, a 147 bp DNA fragment wrapped around an octamer of histone proteins, are considered to be the most basic of the chromatin structure. At an intermediate level, the chromatin is organized in protein-mediated DNA loops that facilitate the interaction between genomic loci that lay distant in the linear genome, many of them linking *cis*-regulatory elements to their target genes. At a higher level of chromatin organization, DNA is organized into topologically associated domains (TAD) in which multiple interactions are favoured, while limiting contacts with genomic regions positioned outside the same TAD [73]. At a mesoscale, chromatin is further organized into functionally distinct megabase large compartments, i.e., A/B compartments. Compartments are enriched for gene-rich and open chromatin regions, while B compartments are gene-poor and closed [74]. The compartmentalization of the genome ensures proper gene expression patterns and regulatory circuits in the normal cell state, and thus perturbations of the 3D genome may have an effect on the gene regulatory network of cancer cells [75]. 

Architectural proteins such as CTCF and SMC-family complex, cohesin, are known to play a critical role in the 3D organization of the genome, and several studies have demonstrated that the genetic or epigenetic alteration of CTCF binding can lead to disruption of 3D genome organization and may result in aberrant gene expression. Both CTCF and cohesion-binding sites suffer an exceptionally high mutational rate in cancer genomes [76,77,78]. Canela and colleagues demonstrated that TOP2B, which is recruited by CTCF to relieve torsional stress by inducing DNA breaks in response to replication or transcription, could also create DNA lesions at loop anchors. Notably, the authors showed that these DNA lesions frequently occurred at breakpoint clusters commonly translocated in human cancer [79]. Another study by Kaiser et al. showed that functional CTCF-binding sites at loop anchors and TAD boundaries were prone to mutations that had a functional impact on higher-order chromatin and aberrant cancer gene expression [78]. Recurrent microdeletions in acute lymphoblastic leukaemia have been reported to remove CTCF-associated boundaries in the TADs, allowing proto-oncogene activation [80]. By interrogating the 3D genome organization of primary human leukaemia specimens, Kloetgen revealed widespread differences in intra-TAD chromatin interactions and TAD boundary insulation and specifically identified a TAD “fusion” event linked to the absence of CTCF-mediated insulation that enabled direct interaction between the MYC promoter and a distal super-enhancer [81]. CTCF binding can also be influenced by epigenetic modifications, including DNA methylation. Indeed, CTCF occupancy is inversely correlated to DNA methylation of its binding sites [82,83], and this pattern resulted was altered in multiple cancer settings [32,84]. 

Alterations in local chromatin interactions are a hallmark of cancer genomes and have frequently been associated with activation of oncogenes and silencing of tumour suppressor genes. Recent studies have shown that structural variations (SVs) can modify the 3D chromatin structure by disrupting regulatory sequences that control higher-order chromatin organization [85]. Recurrent somatic SVs have been linked to ‘enhancer hijacking’, whereby proto-oncogenes relocate in close proximity to active enhancers and initiate oncogenic activity [86,87]. The 8q24 region, which harbours a cancer-associated risk variant and multiple enhancer elements, has been shown to directly interact with the MYC proto-oncogene via long-range interactions and to regulate its expression in various cancer types including colorectal, prostate, breast, lung, and leukaemia [88,89,90,91,92]. SV can also alter the copy number of regulatory elements. For instance, the increased copy number of the enhancer located 650 kb centromeric to the androgen receptor (AR) gene has been linked to the activation of the AR gene, leading to increased proliferation and decreased sensitivity to enzalutamide [93]. A recent study, investigating 288,457 somatic structural variations (SVs) in 2658 cancers across 38 tumour types, concluded that while SVs can lead to the fusion of discrete TADs and complex rearrangements, only 14% of the boundary deletions resulted in a change in expression, thus once again indicating that gene regulation in (cancer) cells is complex and multifactorial [94]. In this regard, recent data indicate that the patterns of somatic structural variation greatly vary across human cancer genomes [95], yet whether a higher level of SV mutation burden is associated with a more widespread rearrangement of the 3D cancer genome is still to be deciphered. We anticipate that future longitudinal studies focusing on the evolution of 3D cancer genomes during carcinogenesis across different tumour types will shed light on the relationship between genome alterations and 3D genome organization.

### 2.3. Epigenetic Heterogeneity in Metastasis

As described above, epigenetic changes induced by extra- or intracellular factors can prime cells for the successful evasion from the primary site favouring the onset of disseminated tumour cells (DTCs) that can colonize distal sites to form metastatic lesions. After successful dissemination, DTCs gain cell plasticity, favouring their survival by an immune escape, to ultimately support metastasis overgrowth. For cancer cell homing, cells can reactivate development-associated transcriptional programs, which allow them to adapt to the new environment. This was shown for castration-resistant metastatic prostate cancer, in which enhancer hijacking and reactivation of developmental pathways resulted in successful homing of metastatic clones [96]. In another approach, Makohon-Moore and colleagues sequenced 26 PDAC metastases to identify the mutations driving the individual metastasis formation [97]. Intriguingly, they came to the conclusion that most of the mutations were common between the samples, although the phenotypic characterization of the tumours showed clear differences. These results pointed to additional drivers of metastasis, probably by changes in the epigenome, resulting in an increase in eITH. In the same cancer type, McDonald and colleagues could show that epigenetic reprogramming is a major driver of distant metastasis, as the presence of histone marks changed between local and distal metastases at large organized chromatin histone H3K9-modified (LOCK) heterochromatin domains [98]. These regions showed features of repressed chromatin in the primary tumours but were reactivated in the distant metastasis, driving the upregulation of genes important for metabolic adaptation to the new environment. Moreover, it has been shown that the activation of metastatic-specific enhancers supported invasion and tumour growth in a foreign microenvironment [99,100]. Of importance is that the deletion of metastatic-specific enhancers reduced the expression level of the regulated genes, impairing the metastatic capability of the targeted cells. These results highlighted the pro-oncogenic activity of single-regulatory elements as cell-specific drivers of cancer progression. Further studies should elucidate the biological relevance of these findings by dissecting the mechanisms by which epigenetic reprogramming at enhancers is put in place and maintained within the metastatic cell population. It would be of high relevance to understand whether epigenetic reprogramming occurs already in the early DTCs and to unveil their relative level of heterogeneity by applying single-cell approaches.

## 3. Measuring Epigenetic Heterogeneity: From Population Analyses to Single-Cell Omics

Cancer samples consist of a complex and heterogeneous network of cellular interactions that govern tumour progression and metastasis formation. These complex ecosystems have traditionally been profiled using bulk experimentation studies that have only revealed an averaged cellular behaviour over thousands of cells, thus veiling the real intra-tumour heterogeneity and subsequently limiting our understanding of the biological implications and molecular mechanisms driving such functional cancer heterogeneity. The advent of single-cell transcriptomics and epigenomic methods has facilitated a comprehensive analysis of the various cellular programs and mechanisms of genetic regulation within heterogeneous cell populations at the single-cell level. In this section, we focus on recent evidence of the cell-to-cell genetic and non-genetic tumour heterogeneity and its implications on tumour growth, evolution, metastasis, drug resistance, and therapy, among others.

### 3.1. Unravelling Intratumour Heterogeneity (ITH) by Single-Cell Omics

#### 3.1.1. Genetic Intra-Tumour Heterogeneity

Application of single-cell genome analyses in cancer research, which can determine whether mutations are in the same or different cells, confirmed that individual cancer samples are genetically heterogeneous, as previously inferred from bulk data (Figure 2). Single-cell genomics has provided essential information about the subclonal dynamics of cancer, which in turn has important therapeutic implications, including the evolution of drug resistance (pre-existing clones versus the acquisition of new mutations), characterization of the subclones that can invade and metastasize, or subclonal compositions of the tumour that are associated with clinical outcome. For example, single-cell genome sequencing applied to breast cancer and acute lymphoblastic leukaemia samples before and after treatment revealed that genetically distinct resistant cells within a heterogeneous population of cells were pre-existing and selected during treatment [101,102,103,104,105]. Other single-cell studies demonstrated that mutations enabling tumour progression, metastasis, or relapse may already be present in early preinvasive or even premalignant lesions [106,107,108,109]. In a recent study by Chen et al. [110], the authors performed single-cell DNA sequencing of a sorted stem cell population in longitudinal samples of patients with Myelodysplastic syndrome (MDS) who progressed to myeloid leukaemia (AML) and observed that stem cells at the MDS stage had a significantly higher subclonal complexity compared to blast cells and contained a large number of ageing-related variants. This finding revealed a role of pre-existing rare aberrant stem cells that undergo a pattern of nonlinear, parallel clonal evolution and drive disease progression and leukaemic transformation. Furthermore, a recent not-yet-peer-reviewed study by Williams et al. [111] demonstrated that in myeloproliferative neoplasms, driver mutations were detectable in blood one to four decades before the clinical manifestation of the disease, thus indicating that the acquisition of driver mutations early in life could drive adult blood cancer. However, most non-invasive tumours never progress to metastatic disease, thus suggesting that the presence of rare genetically distinct resistant cells may not be sufficient to drive disease progression. Altogether, these results suggest that genetic evolution is unlikely to represent the only mechanism of cancer progression and resistance to cancer therapies. Hence, non-genetic adaptation of cancer cells is likely to substantially contribute to the intra-tumour heterogeneity that underpins cancer progression and therapeutic evasion, as previously suggested by others [112].

#### 3.1.2. Non-Genetic Intra-Tumour Heterogeneity

The interplay between genetics, epigenetics, and environmental signals sets the gene expression pattern of a cell and thus determines the cell’s fitness and behaviour. Gene expression ITH typically reveals two main layers: very distinct clusters that correspond to different cell-types composing the cancer samples (e.g., malignant cells, T cells, and stromal cells among others) and further subclusters that may represent the specific cellular states, elated to cell cycle, stress, hypoxia, epithelial differentiation, senescence, protein metabolism, and other dynamic programs [5]. scRNA-seq studies on malignant and non-malignant cells indicate that while malignant cells tend to primarily cluster by the tumour of origin, non-malignant cells are typically clustered by the cell type, independently of tumour type [113,114,115,116,117], thus suggesting that inter-individual heterogeneity of malignant cells tends to be larger than that of any given type of non-malignant cell, and higher than the intra-tumour heterogeneity [118]. Recent scRNA-seq experiments have confirmed that tumours constitute a complex and heterogeneous ecosystem, with a subpopulation of cells showing treatment-resistant properties [119], as well as heterogeneous patterns of immune infiltration [114,119,120] and overall cellular composition [12] that are relevant to tumour biology and clinical outcome. Furthermore, although human tumours are shaped by the growth and evolution of driver mutations, single-cell RNA-seq methods have facilitated the identification of developmental hierarchies for malignant tissues in which cancer stem cells give rise to differentiated tumour cells and drive tumour growth [6,107,108]. A recent work by Kinker et al. profiled single-cell expression programs of multiple cancer lines in order to investigate the ITH gene expression in the absence of genetic heterogeneity and a native microenvironment [121]. This study identified a number of expression programs that were recurrently heterogeneous within multiple cancer cell lines, thus demonstrating that cellular plasticity is, at least in part, an intrinsic property of cancer cells.

Epigenetic mechanisms allow for the regulation, maintenance, and inheritance of transcriptional programs. In contrast to genetic mutations, epigenetic changes are dynamic and potentially reversible, leading to heterogeneity during normal development, disease evolution, or in response to environmental stimuli. Loose epigenetic constraints in cancer cells are likely to enhance cellular plasticity and allow switching between cellular states, thus favouring cancer cell adaptability and resistance to therapy. Pioneering work by Corces et al. [122] demonstrated cell-to-cell tumour heterogeneity of leukaemic cancer regulomes, measured by chromatin accessibility, was associated with variations in the expression of the associated genes, thus indicating a functional link between accessibility and gene expression variability. Furthermore, the authors found that the majority of cancer cells within a patient were clonally derived and harboured all the leukaemic mutations at comparable allele frequencies, suggesting that the epigenomic heterogeneity is somewhat independent of the genetic heterogeneity of leukaemic cells [122]. Longitudinal single-cell transcriptional, epigenetic, and genetic cancer studies have also demonstrated that the genetic subclonal structure in cancer cells can be further diversified by epigenetic memory, thus increasing the plasticity and heterogeneity of transcriptional programs [123] and generating phenotypically distinct subclones that are differentially responsive to therapy [124]. Heterogenous epigenomic subpopulations that responded differently to targeted therapy have also been observed in leukaemic cell lines [125]. Epigenomic analysis of sensitive and resistant breast tumour samples at a single-cell resolution showed that the resistant cell population displayed a higher degree of epigenetic heterogeneity than their sensitive counterparts [126]. Specifically, they showed a reduced depositioning of the repressive mark H3K27me3 at genes known to promote resistance to chemotherapy or targeted therapy [126]. Bell et al. found that a proportion of cancer cells pre-existing in the drug-naive tumour population that survived the initial therapeutic challenge were able to adopt a transcriptional state that enabled them to acquire a drug-resistant phenotype [127]. Furthermore, when these cells were withdrawn and then rechallenged, they continued to tolerate the drug in the absence of new coding mutations [127]. This transcriptional plasticity of the malignant cells was associated with enhancer switching phenomena, whereby as cells differentiate, enhancers specific to the naïve state cease their activity and primed enhancers become active. 

#### 3.1.3. Tumour Microenvironment (TME) Heterogeneity

TME can directly interact with tumour cells and influence malignant cell function and thus influence the behaviour of the tumour. Single-cell OMICs experiments have also revealed a certain degree of heterogeneity of the TME both within and between tumours (Figure 2). Heterogeneity of exhausted T cells in the TME was linked to patient survival in hepatocellular carcinoma [128]. Likewise, varying expression levels of cell type-specific markers for various components of the TME, including the cancer-associated fibroblasts, endothelial cells, and infiltrating immune cells, were significantly associated with the survival of patients with pancreatic ductal adenocarcinoma [120], while glioblastoma tumours showed a highly heterogeneous immune cell infiltration, with patients with mixed myeloid and T-lymphoid infiltrates showing worse outcomes [129]. The composition of the TME has also been reported to modulate the therapeutic response. Analysis of immune cells in the HCC microenvironment using sc-RNA-seq revealed three distinctive HCC subtypes with immunocompetent, immunodeficient, and immunosuppressive features [130], which are likely to have an impact on tumour response to therapy. Recent sc-RNA-seq studies have demonstrated an association between the responsiveness to immune checkpoint inhibitors and the presence of tertiary lymphoid structures and clonal B cell expansion in melanoma [131]. An independent study has identified novel MHC class II-expressing cancer-associated fibroblasts capable of presenting antigens to CD4+ T cells, with the potential to modulate the immune response in pancreatic tumours [132]. Furthermore, Ebinger et al. identified a rare subpopulation of cells with combined properties of long-term dormancy, treatment resistance, and stemness in acute lymphoblastic leukaemia patients (ALL) [105]. They further determined that dissociation of the resistant cells from the in vivo environment made them sensitive to treatment, thus highlighting the clinical implications of the TME heterogeneity [105]. These findings suggest that some cancer patients might benefit from therapeutic strategies involving the release of resistant cells from the niche. Potential limitations of the above-mentioned sc-OMICS strategies to dissect TME heterogeneity are the loss of spatial information of the cells within tumour tissue as well as the introduction of spurious variations in gene expression profiles due to arduous processing of the solid tumour issues. In this regard, a number of imaging technologies have been developed that allow generating spatially resolved single-cell transcriptomic data [133,134,135] (Figure 2). For example, the application of multiplex imaging to invasive tumours has identified clusters of PD-L1+ macrophages at the margin of the tumours, thus suggesting their potential contribution to modulating T cell entry in the tumour lesion [136]. Future efforts using multiparametric imaging, mass cytometry, and single-cell RNA sequencing will improve our understanding of the association between the cellular composition, functional status, and spatial distribution of the TME and relevant patient outcomes. In sum, although multiple studies have highlighted the contribution of the epigenetic changes to eITH and their importance in guiding tumour progression and drug resistance, there are still relevant open questions that need to be addressed in the next future.

Key points: Tumour eITH plays a key function in tumour progression and metastasis.Genomic heterogeneity plays an important role at the onset of tumorigenesis, which is later driven by non-genetic changes.Proper dissection of the tumour at the epigenetic level allows for personalized therapeutical approaches and prediction of tumour burden.The development of single-cell approaches facilitates the identification of tumour subpopulations.Single-cell OMICs has revealed a far larger clonal complexity of the tumours than previously expected.Our increasing understanding of the subclonal dynamics of cancer and its diverse trajectories using single-cell omics has enabled a better characterization of mechanisms of drug resistance and metastasis that will offer innovative therapeutic opportunities.Epigenetic alterations can further diversify the genetic subclonal structure of the tumours and generate phenotypically distinct subclones with differing therapeutic resistance, thus underscoring the importance of targeted epigenetic drugs for future combination therapies.Analysis of the TME at the single-cell resolution indicates that the composition of the TME, as well as its interplay with the tumour cells, can also modulate the therapeutic response.Imaging approaches allow for the spatial resolution and the analysis of the interplay between tumour and TME.

Challenges:Identification of the main epigenetic mechanisms driving resistance, relapse and metastasis burden.Linking phenotypic features and molecular characterization of the resistant and metastatic cells.Spatially and temporally resolved single-cell studies to better understand the role of TME in tumour evolution and leveraging this information for better combination therapies.Understanding the relationship between genomic and epigenomic alterations in cancer cells.The translation of single-cell omics into the clinical routine is still hampered by their limited high-throughput, relatively high costs, and lack of technical and computational standarized procedures.

## 4. Concluding Remarks

Characterization of tumour subpopulations and their epigenetic state is still a major challenge of today’s research and therapy development. Evidence suggests that tumours and the surrounding microenvironment comprise a heterogeneous mixture of different subpopulations, as shown by single immunofluorescence imaging and more recently by sc-OMICs approaches. Recent studies have shown that changes in the genome, epigenome, and transcriptome of rare subpopulations, all of which were masked and undetectable by bulk genomic analysis beforehand, can drive cancer growth, progression, and drug resistance and thus have a profound implication in cancer biology. Specifically, cancer epigenomic studies at single resolution have come to demonstrate that epigenetic heterogeneity within individual cancer samples is functional, further increases the plasticity of transcriptional programs, and thus influences drug sensitivity and clonal dynamics of cancer evolution. Moreover, recent developments in spatially resolved single-cell genomics promise to help us to unravel the spatio-temporal distribution of epigenetic and phenotypic features in single-cells and will thus improve our understanding of the association between cellular composition and spatial distribution of the tumour and surrounding microenvironment cells. This knowledge can be translated into the clinic by providing customized treatment options for cancer patients, targeting the different tumour subpopulations identified.

## Figures and Tables

**Figure 1 cancers-13-04969-f001:**
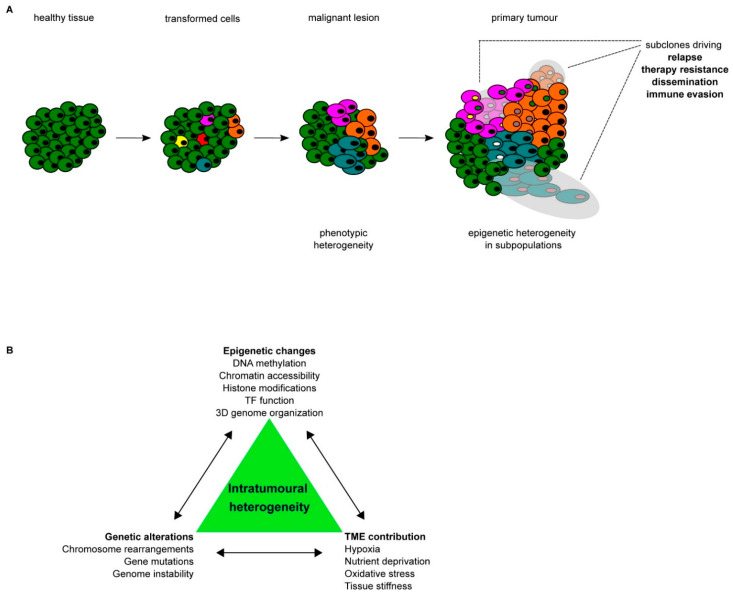
Tumour development; (**A**) schematic representation of the stepwise progression from healthy tissue towards a primary tumour, including the increase in heterogeneity on both the phenotypic and epigenetic levels. Different colors of cells indicate different subpopulations within the same tumour bulk, whereas different colors of nuclei indicate epigenetic heterogeneity within the same subpopulation of cancer cells. (**B**) Schematic summary of the different drivers of intra-tumoural heterogeneity (ITH) and their interconnectivity.

**Figure 2 cancers-13-04969-f002:**
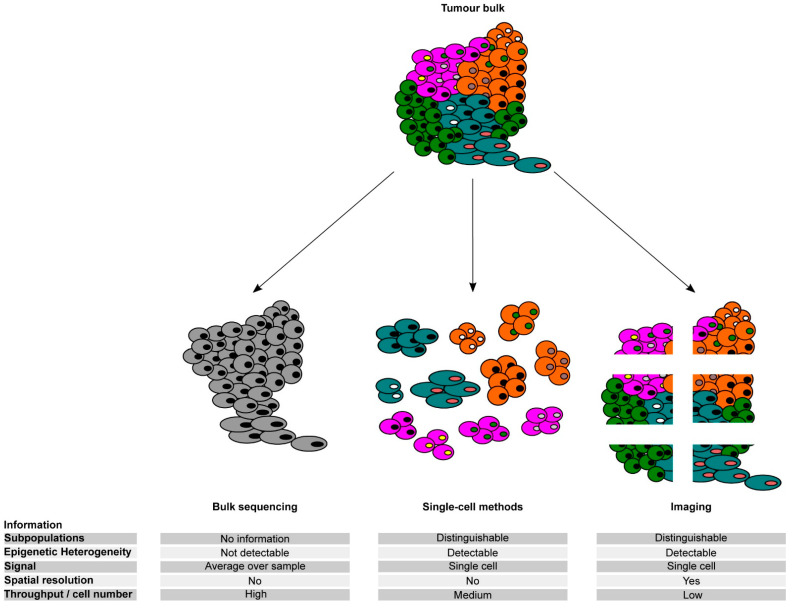
Comparison of different methods for untangling the epigenetic heterogeneity in cancer. Schematic representation of the potential of bulk, single-cell, and imaging approaches to address phenotypic and epigenetic heterogeneity in tumour samples and their main characteristics.

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
