# Peer review of "An Epigenetic Perspective on Intra-Tumour Heterogeneity: Novel Insights and New Challenges from Multiple Fields"

_cancers, 2021, doi:10.3390/cancers13194969_

Round 1

Reviewer 1 Report

The review paper by Zippo et. al. provides a nice and rather comprehensive review of what is known about epigenetic intra-tumor heterogeneity.

The paper would be strengthened and likely more cited if the authors consider the following for revision.

  1. Make a table of the key papers and their conclusions-its in the manuscript but the readers would likely appreciate it having a table.
  2. List what are the novel insights.
  3. List what are are the new challenges
  4. At the end list Key Points
  5. There are minor grammatical and stylistic errors - e.g. the use of the word Albeit is not in the correct context 

Author Response

We agree in principle with the suggested changes proposed by Reviewer #1 and added a list of the novel insights and upcoming challenges in the revised version of the manuscript (from lane 587 to 627). We also carefully checked the manuscript for any grammatical or stylistic errors, as requested. However, we think that adding a table with the key papers may be limiting the breath of this review as the indicated papers have a specific added value, with respect to the context in which they are cited. For these reasons we would prefer to not “weight” the relative contribution of few papers by including a specific table, yet we prefer underlying their relevance within the text of the manuscript.

Reviewer 2 Report

In this review manuscript, the authors first describe the major epigenetic mechanisms, including DNA methylation, histone modifications, and chromatin remodeling, that are involved in human tumorigenesis. Then, they discuss how single-cell based approaches are contributing to dissect the key role of epigenetic changes to tumor heterogeneity. They also highlight the importance of dissecting the interplay between genetics, epigenetics and tumor microenvironments to understand the molecular mechanisms governing tumor progression and drug resistance. This review is well-written and provides a relatively comprehensive summary of current knowledge and information of epigenetic perspective on intra-tumor heterogeneity. Here are my comments.

  1. Despite recent progression of single-cell epigenomic methods, there are still challenges to routinely measuring the epigenomic changes at the single cell level, particularly in clinical tumor samples. It is better to discuss these technique challenges in the manuscript.

  1. What is the translational potential of single-cell OMICs approaches?

  1. Page 13 Line 400: the “castration-resistant metastatic pancreatic cancer,” or prostate cancer.

  1. Page 13 Line 535-538: Please add reference for the statement: “Analysis of immune cells in the HCC microenvironment using sc-RNA-seq, revealed three distinctive HCC subtypes with immunocompetent, immunodeficient, and immunosuppressive features. The composition of the TME has also been reported to modulate the therapeutic response.”

Author Response

We appreciated the general comments from Reviewer #2 regarding the relevance of this manuscript. We also addressed his/her comments and commented on the current challenges related to single cell epigenome profiling on clinical tumor samples. On the same line, we commented on the translational potential of applying multi-Omics approaches to dissect ITH from lanes 582 to 627 of the revised manuscript; lanes 646 to 652).

We also corrected the mistake in line 400, referring to pancreatic and not prostate cancer, as requested. Finally, we included a reference to support the statement of page 13 (lines 535-538).